# Classification of Malware Families Based on Efficient-Net and 1D-CNN Fusion

Xulei Chong [1], Yating Gao [2], Ru Zhang [1,*], Jianyi Liu [1], Xingjie Huang [2] and Jinmeng Zhao [2]

1   School of Cyberspace Security, Beijing University of Posts and Telecommunications, Beijing 100876, China
2   State Grid Information & Telecommunication Branch, Beijing 100761, China
*   Correspondence: zhangru@bupt.edu.cn

**Abstract:** A malware family classification method based on Efficient-Net and 1D-CNN fusion is proposed. Given the problem that some local information of malware itself as one-dimensional data will be lost when the malware is imaged, the malware is converted into an image and one-dimensional vector and then input into two neural networks. The network of two-dimensional convolution architecture is used to extract the texture features of malware, and the one-dimensional convolution is used to extract the features of local adjacent information, the deep characteristics of different networks are fused, and the two networks are modified at the same time during backpropagation. This method not only extracts the texture features of malware but also saves the features of the malware itself as one-dimensional data, which shows better performance for multiple datasets.

**Keywords:** deep learning; malware family classification; image classification; feature fusion

## 1. Introduction

Generally, after malware is detected, to analyze it more deeply to obtain important information such as its attack purpose, it is often necessary to classify its family. After determining the family to which the malware belongs, it is convenient for malware analysts to conduct a more targeted analysis. Especially in today's era of big data, the number of malware families has exploded [1].

To better analyze and trace the malware, the research on the classification of malware families is still of great significance [2–6]. In recent years, with the success of deep learning in machine vision, natural language processing, and other fields, more and more scholars have begun to combine malware family classification problems with deep learning. The main classes of malware detection techniques are based on static and dynamic analyses. Among them, the static analysis starts with the program file itself, without actually executing it. For example, [7,8] designed a classifier capable of static analyzing malware, and [9,10] designed byte-level features for malware static detection. Different from the essence of the static analysis of malware, dynamic analysis is a method of recording the runtime behavior of malware when running it and extracting the signature, which generally needs to be executed in a controlled virtual environment. In addition, [11,12] used a file, registry, and network activity information to train dynamic analysis malware classifiers.

However, there are several obvious dilemmas with these approaches. On the one hand, the malware itself is not 2D data, and by transforming it into an image of 2D data, unnecessary priors are introduced [13]. On the other hand, when generating malware images, it is necessary to specify an additional hyperparameter, that is, the width of the image for tuning. The specified width often means unpredictable truncation of the malware, and this truncation may cause the original contextual malware to be truncated, which may lead to the loss of local features of some malware.

In view of the above issues, this paper proposes a malware family classification method based on Efficient-Net and 1D-CNN. The main work of this method includes the following three points:

1. To avoid the information loss caused by the conversion into images, we convert the binary files of malware into one-dimensional vectors at the same time and use one-dimensional convolution to extract the features of local adjacent information.
2. To improve the accuracy of classification, we introduce Efficient-Net with good quantitative adjustment ability to obtain the best network parameters conducive to malware classification, and we integrate the texture features and adjacent features of malware to discriminate malware categories from multiple angles.
3. To adapt to different lengths of malware, we also discuss the setting of the image width and extract an adaptive image width setting method.

The rest of this paper involves the following parts: Section 2 introduces representative malware detection methods. The overall framework and design details of the model proposed are described in Section 3. Section 4 is indicated the comparison results and analysis through experiments. Ultimately, the full text is summarized in Section 5.

## 2. Related Work

Moskovitch et al. [7] first proposed an opcode-based machine learning method to detect unknown malicious files and used a variety of classification algorithms to classify the opcode feature set. Later, Shabtai et al. [8] extended the research work of Ref. [7], divided the opcode sequence into four different sizes, and used multiple classifiers for classification, and the classification accuracy reached 96%. The opcode-based method is time-consuming to extract features. Later, Kim et al. [14] proposed a technique for malware detection based on PE headers, which improved the efficiency. However, this technique will not work well if the original PE header of the file is obfuscated. Besides opcodes, the API of PE files and their system calls are also very distinguishing features. Research has shown that API calls can be used to model the behavior of programs. Essentially, API functions and system calls are related to services provided by the operating system, and accessing system resources means calling APIs, so calls to specific APIs provide key information that represents the behavior of the malware. However, it is very difficult to extract the API calls of the obfuscated malware, and this method has a high false-positive rate. There are also some studies using lower-level byte-level features for malware detection [9,10], but they are still inherently vulnerable to obfuscation attacks.

Earlier research focused on features based on API call sequences; Rhode et al. [15] collected API call sequences and machine metrics through a sandbox and then fused the two feature vectors into a single vector using neural networks, random forests, and (Support Vector Machine) SVM as a classifier to detect malware. There is also a combined machine learning framework proposed by Lu et al. [16], which extracts API sequence features and uses the random forest and recurrent neural network processing to process them separately. Finally, the two methods were combined. The detection accuracy of the method reached 99.3%. In some recent studies, researchers have turned their attention to in-memory; Yucel et al. [17] proposed a technique based on an in-memory image of an executable file, they used a virtual machine to execute a malware sample and created a 3D image of the memory, using the similarity calculation between 3D images and known malware to detect malware. In addition, there are some studies on exploiting the network traffic of malware [18,19] for detection.

At present, there are relatively few anti-obfuscation studies on malware detection at home and abroad. An earlier study [20] used a dynamic taint analysis method to record the runtime behavior of the target to build a signature library and optimize the signature library to better identify obfuscated malware. The use of dynamic analysis is a path-dependent-driven obfuscated malware detection method proposed by An et al. [21]. This method uses ISR for dynamic debugging and drives malware by solving the constraints of path conditions during the debugging process. The code executes different paths for deeper detection to hide malware. In addition, there are also studies [22] that improve the problem that the N-gram feature extraction algorithm is easily affected by obfuscation operations and realizes the removal of obfuscated features by dynamically calculating the obfuscation

threshold in different malware samples. With the development of malware visualization technology, some research [23] proposed the construction of anti-aliasing image texture features. The fusion feature of interference solves the problem that the accuracy of global feature classification decreases sharply when the grayscale image of malware has a high similarity or large difference. However, the method based on dynamic analysis is time-consuming and has the problem that the execution logic of malware cannot be completely covered, and some malware will detect that it is executing in the sandbox, thereby hiding its malicious behavior. The existing classical malicious code analysis methods are shown in Table 1.

**Table 1.** Summary and comparison of existing methods.

| Class | Ref. | Method | Characteristic | Disadvantage |
|---|---|---|---|---|
| Static analysis | [7] [8] [14] [9,10] | An opcode-based detection Multiple classifiers for detection A PE headers-based detection Low byte-level features for detection | Using multiple classifiers to detect the opcode feature set. Extracting features from PE header files and API files | They are difficult to extract the API calls of the obfuscated malware and have a high false-positive rate. |
| Dynamic analysis | [15] [16] [11,12] [17] [20] [23] | API call sequences and machine metrics Combined machine learning framework Network activity info to train classifiers A method based on the in-memory images of executable files A dynamic taint analysis method A method based on constructing anti-aliasing image texture features | Using machine learning for detection  Biased towards in-memory  Anti-obfuscation research | Time-consuming, the execution logic of malware cannot be completely covered |

As mentioned above, on the one hand, the existing static analysis methods are simple and easy to implement, and they extract low-level features for static detection of malware. However, they have difficulty extracting the API calls of the obfuscated malware, as well as a high false-positive rate. On the other hand, the existing dynamic analysis methods make full use of machine learning knowledge, which is more conducive to malicious file detection and de-obfuscation in memory. However, they currently introduce unnecessary priors when the malware is imaged, and their imaging hyperparameter also results in the truncation of the malware unpredictably by imaging 2D images. This truncation causes the loss of local features of some malware. To alleviate this dilemma, the proposed method integrates the deep features of the 2D convolutional network and the 1D convolutional network; 1D vectors can compensate for the loss of functional features in 2D images, with the advantage of uniform scaling with other neural networks. This can compensate the problem of feature loss caused by malicious image truncation.

### 3. The Proposed Method

This paper proposes a malware family classification method based on Efficient-Net and 1D-CNN to reduce these dilemmas. The overall architecture of the method is shown in Figure 1.

From Figure 1, we can clearly see the work flow. The input of the whole architecture is the PE file of malware, and the output is the extracted malicious code detection feature. Firstly, the input PE file is mapped into a 2D gray image and a 1D sequence, so as to extract their features later; the 2D gray image is converted according to the corresponding gray value of the PE file. Then, the transformed 2D gray image uses the efficient-Net with pyramid feature fusion to extract features of different dimensions so that the location and detail information of the low-level features and the semantic information of the high-level features are fused. For the 1D feature, a simple and efficient CNN neural network is used to extract local strong dependencies. Finally, we combine the features extracted by the two methods and use them as the basis for detecting malware.

Next, we will describe in detail the function and working principle of each part.

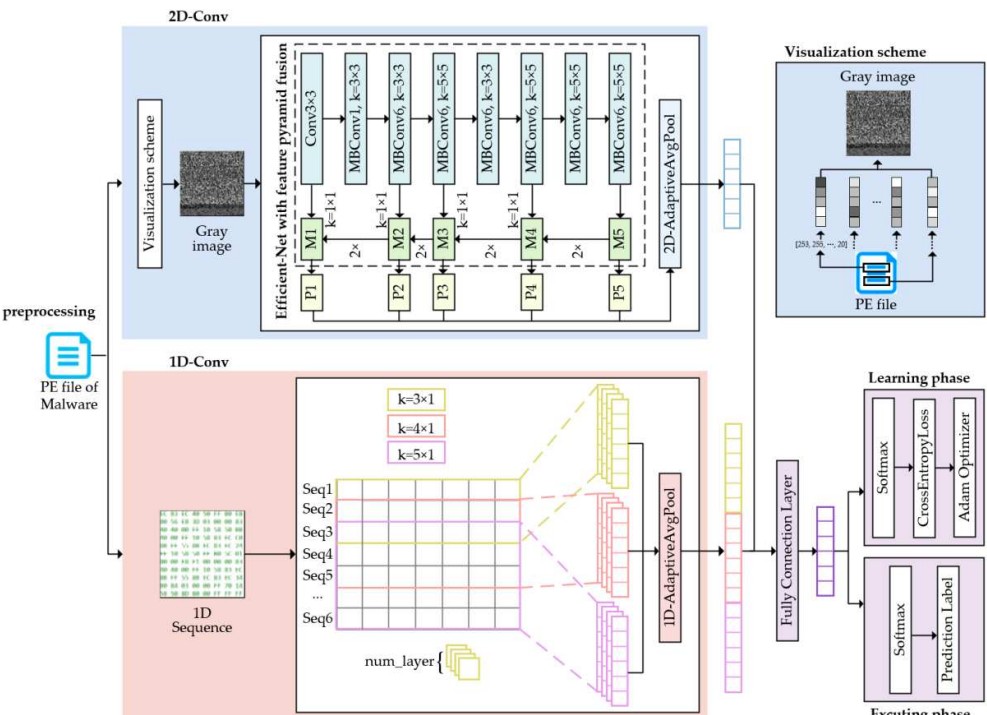

**Figure 1.** Model architecture diagram that combines Efficient-Net and 1D-CNN.

### 3.1. Malware File Preprocessing

According to statistics, the current Windows PE files are the most common malware types on web sites [3], so this paper focuses on the classification of Windows PE files for malware families.

The main fields of the PE file include fields such as .text, .data, .rdata, .idata, and .edata. The .text field is a code segment, which saves the actual code of the program. This is the only code segment in the PE file, and the other fields are data segments. In the data segment, the .data field saves important information such as the global variables and global constants of the program. The .rdata field is the resource data segment, the type, size, and location of the file that the PE file will use during the running process. The .idata field stores the address of the imported function and the external function. Similarly, the .edata field stores the address of the exported function, but it is not commonly used and will not be introduced here.

Because the PE header has lower amounts of useful information, the PE header is easily attacked by means such as obfuscation and encryption. This article extracts the detailed information of the field part of the PE file and uses this information to visualize the malware PE file. To extract these field data, it is necessary to disassemble the PE file to obtain its corresponding .asm file. This article uses the IDA pro tool. It can be seen that the .asm file disassembled by IDA pro has detailed information for each field. After extraction of the data of each different field, the method of adaptively setting the width of the malware image in this article is used to visualize the malware. After being converted into an image and the 1D sequence, it can be used to study the problem of malware family classification; the equation of the 1D sequence initialization is as follows.

$$\text{Seq} = \text{random}(0,1) \tag{1}$$

where Seq represents the 1D sequence converted by the malware family.

### 3.2. Width-Adaptive Malware Visualization Scheme

After obtaining the .asm file corresponding to the malware PE file, we can use it to visualize the malware and obtain the grayscale image of the malware. A grayscale image

is an image with a channel number of 1. Its storage method is a two-dimensional matrix. Each value in the matrix corresponds to a pixel in the grayscale image, and its value range is 0~255; 0 means full black, and 255 means full white. The value in the middle indicates that the color of the pixel gradually becomes lighter from black to white. Based on this, we know that the value of each pixel is between 0 and 255, which corresponds to 8 bits in binary, that is, one byte. To convert the .asm file into a corresponding grayscale image, we can read the asm file in binary mode and regard each byte of data in its binary string as a pixel in the grayscale image, so that first the conversion of binary data to pixels is completed. However, in this way, what we get is just a vector of pixel data arranged in one dimension. We also need to set a hyperparameter, that is, the width W of the image, and take the one-dimensional vector just obtained as a row for every W pixels to obtain a converted grayscale image of the malware.

In the above-mentioned malware visualization process, the hyperparameter W needs to be set manually based on experience. However, the file sizes of different types of malware are often quite different. The small ones are only a few KB, and the large ones are dozens of MB. If the width is set fixedly, the shape of the grayscale images of different types of malware will be too different, thus affecting the final classification accuracy. Therefore, Ref. [4] proposes a fixed-width method for malware files of different sizes. However, this method is somewhat rough for today's neural networks that are more sensitive to local details (see Ref. [4]). When this method was proposed, the neural network algorithm was not used for subsequent malware family classification. Today, the application of deep learning is more extensive. In this paper, we will discuss a more suitable method for adaptively setting the width based on the characteristics of image scaling during image classification and will also discuss in the experimental part. Because of the neural network of image classification, it is generally necessary to scale the image to the same length and equal size for subsequent processing. The process of a width-adaptive malware visualization scheme proposed in this paper is shown in Algorithm 1.

---

**Algorithm 1:** Width-Adaptive Malware Visualization Scheme

---

**Input**: Malware asm file.
**Output**: Grayscale image of malware.

---

1: Count the number of segments $N$ of the asm file;
2: Initialize the one-dimensional array Seg that stores the number of bytes of each segment, and initialize the total number of bytes $S = 0$;
3: **for** $i$ in $0, \ldots, N-1$ **do**:
4:     $\mathrm{Seg}_i$: The number of bytes of the $i$-th segment;
5:     $S = S + \mathrm{Seg}_i$;
6: $W = \lceil S \rceil$;
7: **if** $W \times W = S$ **then**:
8:     Jump to line 14;
9: **else**
10:     **for** $i$ in $0, \ldots, N-1$ **do**:
11:         Calculate the number of padding bytes required for the current segment $W_i = W \% \mathrm{Seg}_i$;
12:         Padding the end of segment $i$ with $W_i$ zero bytes;
13: **end if**;
14: Initialize the matrix corresponding to the grayscale image $\mathrm{IMG}_{w \times w}$, $c = 0$;
15: Read the binary stream of asm file $B$, initialization $k = 0$;
16: **for** $i$ in $0, \ldots, W \times W - 1$ **do**:
17:     initialize this pixel $P = 0$;
18:     **for** $j$ in $0, \ldots, 7$ **do**:
19:         $P \leftarrow B_k \times 2^j + P, k \leftarrow k + 1$;
20:     $c \leftarrow (c+1) \% W$;
21:     $r \leftarrow c = 0, r+1 : r$;
22:     $\mathrm{IMG}_{(r,c)} = P$;
23: **end for**;
24: Save the matrix $\mathrm{IMG}_{w \times w}$ as a grayscale image;

---

Since the number of sections often varies between different types of malware, some malware only has .text sections, while other types have all the sections described earlier in this article. For convenience, this article introduces malware with only two fields, .text and .rdata, and other situations are similar. After inputting the asm file, count the number of bytes in the .data section $Seg_0$ and the number of bytes in the .rdata field $Seg_1$ as in the fourth line of the algorithm and then calculate the total number of bytes $S = Seg_0 + Seg_1$. Then, calculate the width to set when it will be imaged, $W = \lceil S \rceil$. At this time, if $W \times W = S$, there is no need for padding; jump directly to line 14 and image the malware with W as the width; otherwise, padding is required. To maintain the integrity of the mapping of the .text field and .radta field in the image to different lines in the picture, they need to be populated separately. Calculate the $W_0 = Seg_0 \% W$ that needs to be filled in the .text field. Similarly, the number of bytes to be filled in the .text field is $W_1 = Seg_1 \% W$, and then perform image processing. During the visualization, since the total number of bytes of each field has been guaranteed to be $W^2$, the 15th line of the algorithm first initializes a matrix $IMG_{w \times w}$ whose long paragraphs are W, and then the algorithm on Lines 17~20 map every 8 bits (one byte) of the binary sequence to the corresponding position in the image.

Compared with the fixed width setting method, the width setting method in this paper has two advantages: first, the malware is fixedly mapped to a square, which reduces the noise caused by scaling because neural networks generally use images. The interpolation algorithm scales the input uniformly, generally a size with equal length and width. With the use of the method in this paper, the malware image will be proportionally scaled by the neural network, which will bring less noise. This paper will further verify and discuss the experimental part. Second, the integrity of the different segments of the malware is preserved, mapping the data of different segments to different rows in the image, rather than having two or more segments in a row.

### 3.3. Convolutional Layer Design

The method designed in this paper includes multiple two-dimensional convolutions and one-dimensional convolution layers. The two-dimensional convolution is used to extract the texture features of malware images. Through two-dimensional convolution, deep abstract features can be extracted; the equation is as follows.

$$\mathbf{F}^{n-1} = \sum_i \mathbf{F}_i^{l-1} * K_i^l + b_i^l \tag{2}$$

$$\mathbf{F}^l = pool(f^n(\mathbf{F}^{n-1})) \tag{3}$$

where $\mathbf{F}^l$ represents the feature extracted by the l-th layer CNN, $K_i^l$ and $b_i^l$ represent the convolution kernel and bias of the l-th layer, pool(.) represents the pooling operation, and $f^n(.)$ represents the activation of the n-th layer. For family classification tasks based on grayscale images of malware, convolutional neural networks can achieve better learning by retaining important features as much as possible and filtering redundant features. The neural network parameters in the two-dimensional convolution structure used in this model are shown in Table 2.

**Table 2.** Partial model parameters of two-dimensional convolution.

| Stage | Input Size | Output Size | Conv Kernel |
|---|---|---|---|
| 1 | $224 \times 224 \times 1$ | $112 \times 112 \times 32$ | $224 \times 224 \times 1$ |
| 2 | $112 \times 112 \times 32$ | $112 \times 112 \times 16$ | $112 \times 112 \times 32$ |
| 3 | $112 \times 112 \times 16$ | $56 \times 56 \times 24$ | $112 \times 112 \times 16$ |
| 4 | $56 \times 56 \times 24$ | $28 \times 28 \times 40$ | $56 \times 56 \times 24$ |
| 5 | $28 \times 28 \times 40$ | $14 \times 14 \times 80$ | $28 \times 28 \times 40$ |
| 6 | $14 \times 14 \times 80$ | $14 \times 14 \times 112$ | $14 \times 14 \times 80$ |
| 7 | $14 \times 14 \times 112$ | $7 \times 7 \times 192$ | $14 \times 14 \times 112$ |
| 8 | $7 \times 7 \times 192$ | $7 \times 7 \times 320$ | $7 \times 7 \times 192$ |

where stage 1 and stage 9 are conventional convolution operations. In stages 2–8, MBConv is used as a basic convolution structure. Figure 2 shows the difference between traditional convolution and convolution using MBConv.

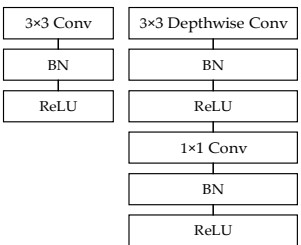

**Figure 2.** Traditional convolution (left), MBConv convolution block (right).

In the MBConv block, the deep-wise convolution and the subsequent $1 \times 1$ convolution are regarded as two independent modules. Deep-wise combined with $1 \times 1$ convolution instead of traditional convolution is not only more efficient in theory, but also due to a large number when using a $1 \times 1$ convolution, this can be done using an optimized math library. Moreover, this separated convolution and the previous convolution prove that the output form is the same.

In addition to two-dimensional convolution, this paper also uses a one-dimensional convolution structure to extract the features of adjacent data. The 1D convolutional structure used is shown in Figure 1. In the one-dimensional convolution structure of this paper, three convolution kernels of different sizes are used, and their convolution sizes are 3, 4, and 5, corresponding to different sizes of receptive fields. In this way, malware with different sizes can be extracted. Correlation features between adjacent data. The number of convolution kernels is set to 100, and then a pooling layer is used to perform pooling operations on different feature maps to extract key data for compressing data and parameters to improve computational efficiency.

### 3.4. Feature Fusion Details

In the feature fusion part of the model, this paper draws on the idea of the Feature Pyramid Fusion Network. First, the feature maps in different convolution layers in Efficient-Net are fused, and then these feature maps are combined with the 1D-CNN volume. The feature maps obtained after the product are fused to obtain a new feature map, which is used for final training and prediction. The equation of this part is as follows.

$$N = \sum_i F_i^{l-1}(X_{(H_i,W_i,C_i)}) \tag{4}$$

where $(X_{(H_i,W_i,C_i)})$ represents the dimension of the input tensor at the i-th layer. The efficient-Net hyperparameter search strategy is also used.

First, the feature maps generated in stages 1, 3, 4, 6, and 8 in Table 2 are selected because the resolution of the feature maps generated in each stage is 1/2 that of the previous stage, and they are features with the same resolution The largest number of channels in the figure (for example, the output feature map of stage 8 is $7 \times 7 \times 320$, the output of stage 7 is $7 \times 7 \times 192$, and the output of stage 8 is selected), which contains richer information. In addition to the feature map of the 8th layer, which is the highest layer, the feature maps of other layers are convolutional with a convolution kernel of $1 \times 1$ to increase the dimension, which is convenient for fusion with the feature maps of the higher layer. The feature maps of higher layers also need to be upsampled to make their resolution the same as the convolutional structure of lower layers. The method of upsampling here is to perform transposed convolution on the feature map. Transposed convolution is an operation that converts a low-resolution feature map into a high-resolution feature map.

Taking the output M4 of the sixth layer as an example, we first need to perform convolution with a convolution kernel size of $1 \times 1 \times 320$ (size $1 \times 1$, number of channels

is 320), and the size is 14 × 14 × 320 after the dimension is increased. The feature map a and then the M5 directly obtained from the feature map of the 8th layer are transposed and undergo convolutional upsampling; the same as in the 14 × 14 × 320 feature map b, M4 is the parametric addition of a and b. At this point, M4 contains the features of its layer and higher layers. Similarly, M1~M3 of other layers can be calculated, which will not be repeated here. In addition, it is necessary to perform 3 × 3 × 256 convolution operations on M1~M5 to eliminate the stacking effect [5] caused by the fusion of different features and scale them to the same size. At this time, we get P1~P5. The other channel also inputs data into 1D-CNN in parallel. After different convolution kernels, different feature maps are obtained, and then a global average pooling layer is scaled to the same dimension to obtain P'1~P'3. The final fusion features are obtained by late fusion of P'1~P'3 in 1D-CNN and P1~P5 generated in Efficient-Net.

Through the detailed description of the above four parts, the malicious code detection features are obtained. Through the calculation of the Softmax classifier and loss function, the feature vector is optimized. The equation for this part is shown as follows.

$$H(p, \overline{p}) = -\sum_{i=1}^{n} p(x_i) \log(\overline{p}(x_i)) \tag{5}$$

$$\theta_{t+1} = \theta_t - \frac{\eta}{\sqrt{v_t} + \varepsilon} \tag{6}$$

where p and $\overline{p}$ represent true and predicted labels, t represents the updated step, $\varepsilon$ represents the learning rate, and $\theta$ represents parameters to be solved for update. In this way, the final trained features are obtained. Next, we will verify the effectiveness of the above models and processes through comparative experiments.

## 4. Experimental Results and Analysis

To verify the effectiveness of the method proposed in this paper, a series of comparative experiments were carried out. Specifically, first the experimental environment, datasets, and hyperparameter settings are described. Then, we conduct experiments on different hyperparameters of the malware family classification method based on the model fusion architecture, explore the influence of different hyperparameters on the experimental results, and determine the optimal hyperparameters. In addition, the method of malware image classification using only a 2-dimensional convolution structure is compared with the model method proposed in this paper. At the same time, to further illustrate the effectiveness of this method, the class activation vector [24] of the feature map is used in this paper. Finally, a comparative experiment was carried out with the methods of other researchers, which further demonstrated the effectiveness of this method.

### 4.1. Experimental Setup

#### 4.1.1. Experimental Environment

The experimental environment of this paper is a server, the CPU configuration is an Intel Core i9-9820k, the physical memory size is 64 GB, and the graphics card is a GeForce RTX 2080Ti with 11 GB video memory.

#### 4.1.2. Experimental Dataset

In this paper, we run the experiment on the VirusShare and MalImg datasets.

The VirusShare dataset consists of 100 G of crawled public samples of malware from the VirusShare website, and for these samples, we filter files except those in the PE format. For files in PE format, we filter and remove samples without .text fields because they have no code segment and cannot be executed, which is equivalent to dead pixel data. We obtained the original malware PE file dataset. For the original PE file dataset, we need to get their labels. This article used Kaspersky Anti-Virus to scan to get the classification information for each PE file. Since we study the problem of fine-grained classification of

Trojans, we screen out the PE samples labeled as Trojans. Further, since each sample has a small label, each malicious sample can be divided into different categories according to the small label. After statistics, due to a large number of samples, there are many kinds of them. There are more than 9000 categories in total. Finally, the categories with several samples greater than 100 are screened out as the final dataset, and there are 38 categories in total.

The MalImg dataset is a public dataset [4]. It is widely used in some malware classification research. This dataset contains malware grayscale images of malware samples of 25 malware families. The number of samples in each family ranges from 80 to 2949, for a total of 9339 samples. The details of the dataset are shown in Table 3. In the dataset, the method proposed in this paper is used for experiments, and the 10-fold cross-validation method is also used for testing.

**Table 3.** MalImg malware image dataset.

| No. | Class | Number of Samples | No. | Class | Number of Samples |
|-----|-------|-------------------|-----|-------|-------------------|
| 1 | Adialer.C | 122 | 14 | Swizzor.gen!E | 128 |
| 2 | Agent.FYI | 116 | 15 | Swizzor.gen!I | 132 |
| 3 | Fakerean | 381 | 16 | VB.AT | 408 |
| 4 | Instantaccess | 431 | 17 | Wintrim.BX | 97 |
| 5 | Lolyda.AA1 | 213 | 18 | Yuner.A | 800 |
| 6 | Lolyda.AA2 | 184 | 19 | Allaple.L | 1591 |
| 7 | Lolyda.AA3 | 123 | 20 | Alueron.gen!J | 198 |
| 8 | Lolyda.AT | 159 | 21 | Autorun.K | 106 |
| 9 | Malex.gen!J | 136 | 22 | C2LOP.gen!g | 200 |
| 10 | Obfuscator.AD | 142 | 23 | C2LOP.P | 146 |
| 11 | Rbot!gen | 158 | 24 | Dialplatform.B | 177 |
| 12 | Skintrim.N | 80 | 25 | Dontovo.A | 162 |
| 13 | Allaple.A | 2949 | \ | \ | \ |

### 4.1.3. Hyperparameter Setup

This paper firstly explores the influence of different output parameters of 1D-CNN in the proposed method on the experimental results of malware family classification. The experimental results are shown in Table 4.

**Table 4.** Experimental results of different parameters.

| Embedding Dim | Acc (%) | The Number of Kernels | Acc (%) | Convolution Kernel Size | Acc (%) |
|---------------|---------|-----------------------|---------|-------------------------|---------|
| 3 | 97.5013 | 3 | **97.5354** | (2, 3, 4) | 97.5183 |
| 4 | **97.5183** | 4 | 97.4585 | (3, 4, 5) | **97.5354** |
| 5 | 97.5100 | 5 | 97.4395 | (4, 5, 6) | 97.5250 |
| 6 | 97.4757 | \ | \ | \ | \ |

Table 4 shows that when the embedding dimension is 4, the classification performance of the model is the best, which is 97.5138%. Subsequently, the embedding dimension of 1D-CNN was fixed, and the effect of different convolution kernels on model performance was also evaluated. When the number of convolution kernels was set to 3, the model performance reached the highest level. The best was 97.5354%. Similarly, after the embedding dimension and the number of convolution kernels was fixed, and experiments were also carried out for different convolution kernel sizes. When the size of the convolution kernel was set to (3, 4, 5), the classification performance of the model was the best. The subsequent series of experiments were carried out based on these hyperparameters. Therefore, the embedding dimension was set to 4, the number of kernels was set to 3, and the convolution kernel size was (3, 4, 5). At the same time, the batch size was set to 32, the learning rate was 0.01, the momentum was 0.9, and the optimizer used Adam; K-fold cross validation re-

quested 10 folds. The malware was filled with values of 0; when this length was insufficient, the longest length in each batch was used.

The method performance indices include the Acc (accuracy rate), P (precision rate), R (recall rate), and F1 (F1-score). The indices are as follows.

$$\text{Acc} = \frac{\text{TP} + \text{TN}}{\text{TP} + \text{FP} + \text{TN} + \text{FN}} \tag{7}$$

$$\text{P} = \frac{\text{TP}}{\text{TP} + \text{FP}} \tag{8}$$

$$\text{R} = \frac{\text{TP}}{\text{TP} + \text{TN}} \tag{9}$$

$$\text{F1} = 2 \times \frac{\text{P} \times \text{R}}{\text{P} + \text{R}} \tag{10}$$

*4.2. Experiments on the VirusShare Dataset*

4.2.1. Experiments with Different Widths of Grayscale Images

To verify the problem that the forcible truncation of malware proposed in this paper will cause the loss of some features of the malware, this paper designed Experiment 1. By setting the hyperparameter of the width of different malware images, the experimental results obtained are shown in Figure 3.

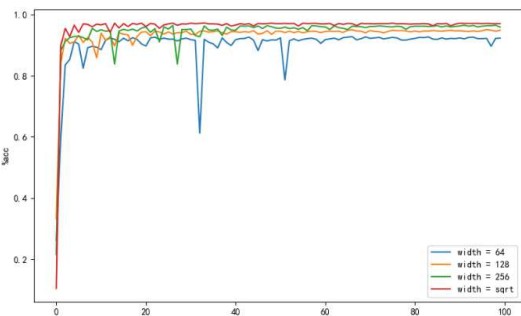

**Figure 3.** Model performance with different widths of malware grayscale images.

As shown in Figure 3, setting different image widths greatly affected the detection performance of the final model. At the same time, this paper also proposed a method for adaptively setting the image width, such as width = sqrt in the above table. Compared with other methods of setting the default width, the model obtained by this method had the best performance. It can be seen that the width adaptation proposed in this paper for the malware visualization method improved the classification performance of the model to a certain extent.

4.2.2. Comparison of Model Fusion

We used 10-fold cross-validation for testing. At the same time, to verify the effectiveness of the efficient-net and 1D-CNN model fusion method proposed in this paper, the performance of the traditional method using only CNN for detection and the method proposed in this paper was compared. As shown in Figure 4, the loss of the method proposed in this paper decreased during training (Figure 4a), which was relatively smooth and did not have some fluctuations that appeared in the comparison method. In the test set loss curve, this phenomenon was more obvious. The figure shows that the curve of the comparison method had an obvious fluctuation phenomenon, which indicates that the model did not converge. The model proposed in this paper converged when the epoch was about 30.

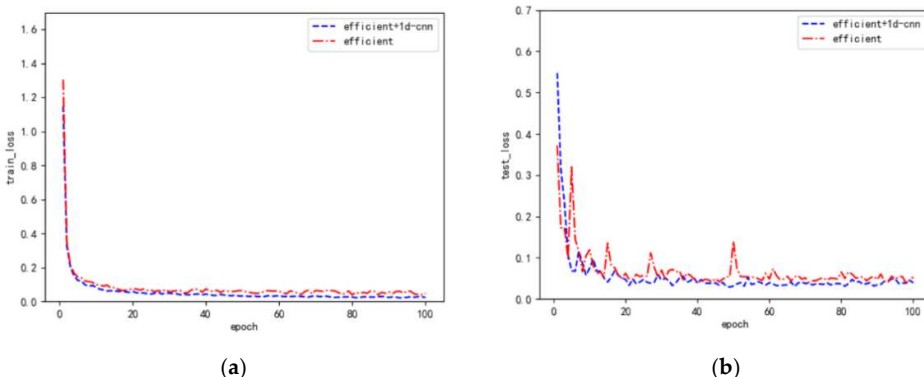

(**a**)                                           (**b**)

**Figure 4.** Comparison of some experimental performance data of the proposed method. (**a**) Loss change of the training set. (**b**) Loss change of the validation set.

The specific results of the experiment are shown in Table 5, i.e., the lowest, highest, and average accuracy in 10-fold cross-validation. The best accuracy rate of the model in this paper was 98.1438%, the average accuracy rate was 97.5354%, and the minimum accuracy rate was 96.9088%. The data of the method proposed in this paper are better than the comparison methods, which shows the effectiveness of the method in this paper.

**Table 5.** Accuracy comparison results of the proposed methods.

| Convolution Kernel | Acc (Max) | Acc (Min) | Acc (Mean) |
|---|---|---|---|
| Efficient net | 98.0938% | 96.8588% | 97.4686% |
| Efficientnet + 1dcnn | 98.1438% | 96.9088% | **97.5354%** |

To further verify the effectiveness of the features extracted by the proposed method, the class activation vector was visualized in this paper, which can effectively show which parts of the feature map have a greater contribution to the classification. The class activation feature heatmaps of some samples of the proposed method are shown in Figure 5. Since the model architecture proposed in this paper fuses two networks and finally performs feature fusion, the weight of the final fully connected layer corresponds to two parts; one part is the weight of the feature map output by Efficient-Net, and the other part is the 1-dimensional convolution The weights of some output feature maps and their sizes are not the same, so their class activation feature heatmaps are output separately. In the example of each sample, the upper part is the class activation feature heatmap output by Efficient-Net, and the lower part is the heat map of the class activation features obtained by 1-dimensional convolution. In the heat map, the darker the color, the more the classification model pays attention to the features there when classifying malware families. From some examples in Figure 5, we can see that when the classification model classifies many samples, in addition to the feature map output by Efficient-Net, the feature map obtained by 1-dimensional convolution also plays a role that cannot be ignored. The heat map shows that the classification model is used for this part. The features are also more involved. As mentioned earlier in this article, this is because 1-dimensional convolution can make up for the loss of some local features caused by the visualization of malware, and the characteristics of 1-dimensional convolution are the features of extracting local data. Therefore, the classifier will also pay attention to the features extracted by the 1-dimensional convolution when calculating the class weight.

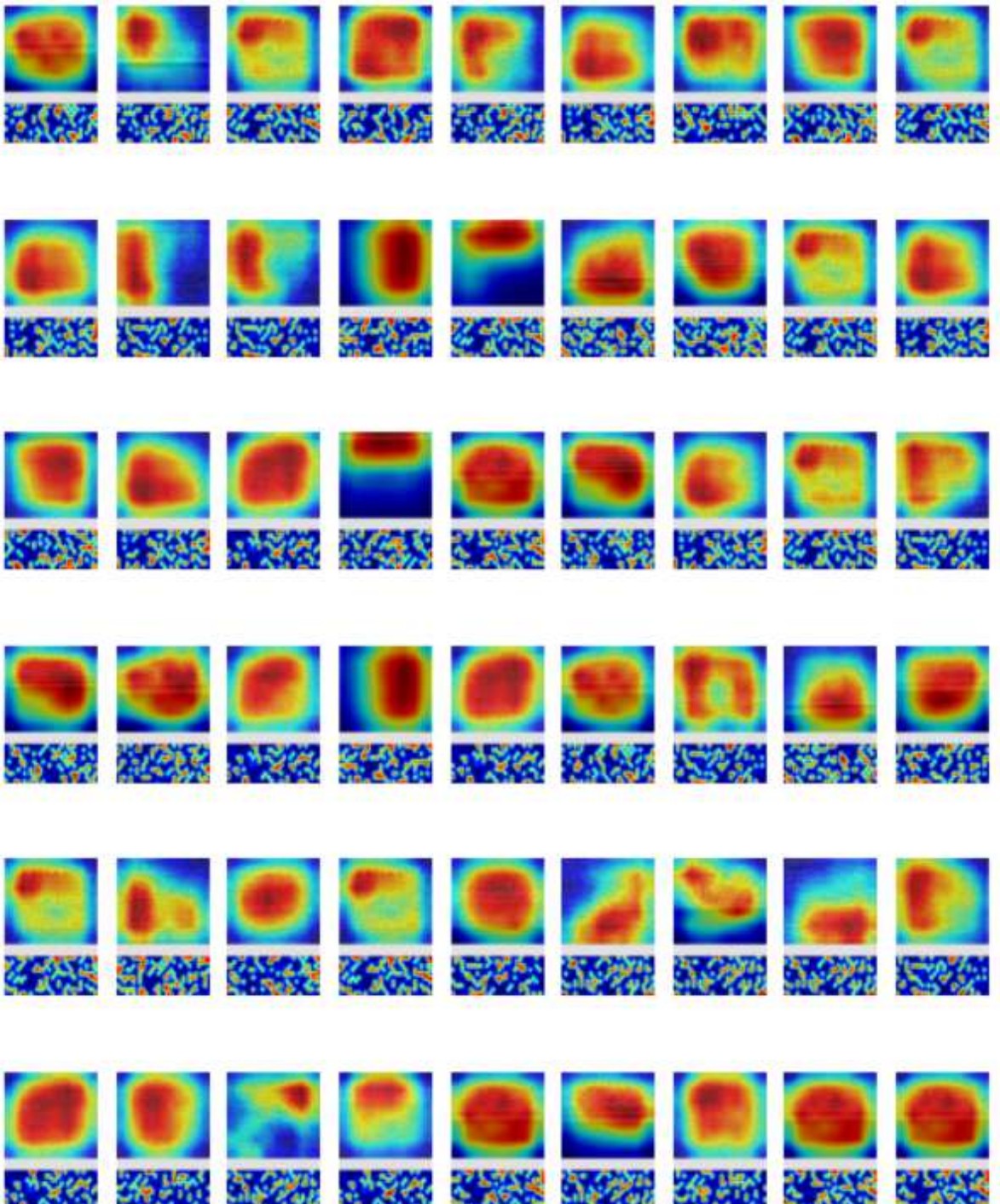

**Figure 5.** Heatmap display of class activation features for some samples.

*4.3. Experiments on the MalImg Dataset*

To verify the effectiveness of the method proposed in this paper, this method was compared with several other methods using this dataset, and the comparison results are shown in Table 6. Compared with most methods, the method proposed in this paper has better classification accuracy.

**Table 6.** Comparison with other scholars' malware family classification techniques.

| Index | Gibert [25] | Cui [26] | Venkatraman [27] | Proposed |
|---|---|---|---|---|
| Acc | 98.48% | 94.5% | 96.3% | **98.55%** |
| R | 96.56% | 94.5% | 91.5% | **97.98%** |
| P | 95.8% | 94.6% | 91.8% | **97.95%** |
| F1 | 95.8% | 94.5% | 91.6% | **97.97%** |
| Preprocessing time | \ | \ | 3 h | **0.5 h** |
| Mean forecast time | \ | \ | 1.5 s | **0.07 s** |
| Time complexity | $O(H \times W \times L)$ | $O(H \times W \times L)$ | $O(H \times W \times L)$ or $O(l^2)$ or $O(N_S^3 + N_S^2 l + N_S d_L l)$ | $O((H \times W \times L)^n)$ or $O(H \times W \times L + l)$ |

where the bold indicates the best performance. "h" and "s" represent hour and second. H and W represent the size of convolutional kernel, and L represents the number of input channels. $N_S$ represents the number of support vectors, $d_L$ represents the dimension of the input vector, l is the number of the training sample, and n represents the number of layers of the feature pyramid. Since the proposed method is a parallel structure, the time complexity is the larger of the two parts. It can be seen that the method proposed in this paper is superior to the existing methods in various indexes such as Acc, R, and so on.

In conclusion, in terms of performance, the proposed method is superior to the best results of other studies, which indicates the effectiveness of the proposed method. In terms of pretreatment time and prediction time, the efficiency of this paper is significantly higher than that of other research, which shows that the method in this paper is more practical.

## 5. Conclusions

Aiming at the problem of unpredictable truncation caused by the setting of the image width after the malware is imaged, the local features of the malware will be lost, and a malware family classification method based on Efficient-Net and 1D-CNN is proposed. This method can compensate for the lack of semantic information. In addition, this paper also proposes a width-adaptive malware visualization method for the width of malware visualization. The experimental results show that the method proposed in this paper is superior to other methods, and the method has the following advantages: (1) The proposed width-adaptive malware visualization method brings less noise and achieves better performance than direct fixed-width methods, avoiding the disadvantages of effective information truncation when processing images. (2) The malware family classification method of the proposed architecture can effectively extract its local features while retaining the texture features of malware. However, the proposed approach has certain limitations. The approach may hardly handle packed malware by advanced packing techniques. In the future, we intend to apply some advanced automatic unpacking/deobfuscation techniques to address this problem. Moreover, the approach is not very effective for classifying unknown malware whose families are not in the training data. We plan to combine unsupervised techniques to improve our proposed approach.

**Author Contributions:** Conceptualization, Y.G. and J.Z.; methodology, X.C. and J.L.; software, R.Z.; validation, J.L.; formal analysis, R.Z.; investigation, X.C.; resources, J.L.; data curation, R.Z.; writing—original draft, X.C.; writing—review and editing, X.C.; visualization, X.H.; supervision, R.Z.; project administration, Y.G. and X.H.; funding acquisition, Y.G. and J.Z. All authors have read and agreed to the published version of the manuscript.

**Funding:** The work is supported by State Grid Corporation of China Headquarters Technology Project, The research and technology for collaborative defense and linkage disposal in network security devices (5700-202152186A-0-0-00).

**Acknowledgments:** The authors would like to thank the anonymous referees for their valuable comments and helpful suggestions.

**Conflicts of Interest:** The authors declare no conflict of interest.

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
