# Peer review of "Classification of Malware Families Based on Efficient-Net and 1D-CNN Fusion"

_electronics, doi:10.3390/electronics11193064_

Round 1

Reviewer 1 Report (Previous Reviewer 2)

Classification of Malware Families Based on Efficient‐Net and 2

1D‐CNN Fusion

1. Introduction section is poorly written and needs major revisions. For example, in-text citation [12-27] without any illustration severely reduces the quality of manuscripts.

2. Flow chart is needed in the methodology section with appropriate notation and clear detailed illustrations.

3. Although table 1 represents the comparisons but more detailed analysis is needed to reflect the noble contributions of your research.

4. I can only see 4 main equations in the manuscript, the readers would like to see the in-depth analysis and flow of values which is not available in the current state of the manuscript. Please improve.

5. Finally conclusion section also needs to be revised.

Author Response

Attached file

Reviewer 2 Report (Previous Reviewer 1)

§   

§  Robust Research

§  This has enough data points to make sure the data are reliable.

§  The author started by describing in simple terms what the data show.

§  Please be concise. Avoid the repeatability of the concept.

§  The results seem plausible.

§  There are sufficient data.

§  The References relevant, recent and readily retrievable.

§  Helpful to the reader.

§  Fair to competing authors.

§  This Gives due recognition to the initial discoveries and related work that led to the work under assessment.

Author Response

Attached file

Round 2

Reviewer 1 Report (Previous Reviewer 2)

Classification of Malware Families Based on Efficient-Net and 1D-CNN Fusion

For comment 1, I still believe that it requires further improvements.

Comment 2 is not addressed by the authors at all only architecture is presented again but flow chart is suggested with standard notations.

Comment 3, more detailed analysis is needed and this section still requires enhancements.

Author Response

One to three pages are the response to reviewer, and starting from the fourth page is the revised and highlighted version of the manuscript.

Round 3

Reviewer 1 Report (Previous Reviewer 2)

Classification of Malware Families Based on Efficient-Net and 1D-CNN Fusion

The current state of the paper may be accepted for publication.

This manuscript is a resubmission of an earlier submission. The following is a list of the peer review reports and author responses from that submission.

Round 1

Reviewer 1 Report

§  This has enough data points to make sure the data are reliable.

§  The author started by describing in simple terms what the data show.

§  Please be concise. Avoid the repeatability of the concept.

§  The results seem plausible.

§  There are sufficient data.

§  The References relevant, recent and readily retrievable.

§  Helpful to the reader.

§  Fair to competing authors.

§  This Gives due recognition to the initial discoveries and related work that led to the work under assessment.

§  The paper's premise interesting and important.

§  The methods used appropriate.

§  The data support the conclusions.

§  The research is publishable in principle and deserves a carefully detailed reading for the reader to find scientific interest and enjoyment.

§  The title properly reflects the subject of the paper.

§  The abstract provides an accessible summary of the paper.

§  The keywords accurately reflect the content.

Reviewer 2 Report

Classification of Malware Families Based on Efficient-Net and 1D-CNN Fusion

I would like to highlight the following suggestions:

1. Introduction part needs major changes, specific contributions should be highlighted pointwise and the organization of the whole manuscript is also missing.

2. Related work section should have a table describing all features and drawbacks with technology used etc.

3. Figure 1 needs revision with more clarity since current form is confusing.

4. What is the time complexity analysis of the proposed framework?

5. Comparison of the proposed framework with other related works mentioned in the literature.

6. Section 4.1 & 4.2 need major changes.

7. Figure 2 needs a detailed illustration with proper mathematical modelling.

8. section 5.2 -feature fusion detail requires deeper analysis with FPN.

9. How your approach is efficient as compared with [9], [10], [11].

10. Result section should have more performance parameters.
